# Associations between Omega-3 Index, Dopaminergic Genetic Variants and Aggressive and Metacognitive Traits: A Study in Adult Male Prisoners

**DOI:** 10.3390/nu14071379

**Published:** 2022-03-25

**Authors:** Francesca Fernandez, Mitchell K. Byrne, Marijka Batterham, Luke Grant, Barbara J. Meyer

**Affiliations:** 1Faculty of Heath Sciences, School of Behavioural and Health Sciences, Australian Catholic University, Banyo, QLD 4014, Australia; francesca.fernandez@acu.edu.au; 2Illawarra Health and Medical Research Institute, University of Wollongong, Wollongong, NSW 2522, Australia; marijka@uow.edu.au; 3School of Medical, Indigenous and Health Sciences, University of Wollongong, Wollongong, NSW 2522, Australia; 4College of Human and Health Sciences, Charles Darwin University, Darwin, NT 0909, Australia; mitchell.byrne@cdu.edu.au; 5Faculty of Engineering and Information Sciences, National Institute for Applied Statistics Research Australia, School of Maths and Applied Statistics, University of Wollongong, Wollongong, NSW 2522, Australia; 6Corrective Services New South Wales, Sydney, NSW 2000, Australia; luke.grant@justice.nsw.gov.au; 7Molecular Horizons, Faculty of Science, Medicine and Health, University of Wollongong, Wollongong, NSW 2522, Australia

**Keywords:** Omega-3 Index, dopaminergic receptors, dopaminergic enzymes, aggression, attention, hyperactivity, genetic polymorphisms, prisoners

## Abstract

Omega-3 long-chain polyunsaturated fatty acids (n-3 LCPUFA) are critical for cell membrane structure and function. Human beings have a limited ability to synthesise docosahexaenoic acid (DHA), the main n-3 LCPUFA required for neurological development. Inadequate levels of n-3 LCPUFA can affect the dopaminergic system in the brain and, when combined with genetic and other factors, increase the risk of developing aggression, inattention and impulse-control disorders. In this study, male prisoners were administered questionnaires assessing aggressive behaviour and executive functions. Participants also produced blood sampling for the measurement of the Omega-3 Index and the genotyping of dopaminergic genetic variants. Significant associations were found between functional genetic polymorphism in *DBH* rs1611115 and verbal aggression and between *DRD2* rs4274224 and executive functions. However, the Omega-3 Index was not significantly associated with the tested dopaminergic polymorphisms. Although previous interactions between specific genotypes and n-3 LCPUFA were previously reported, they remain limited and poorly understood. We did not find any association between n-3 LCPUFA and dopaminergic polymorphisms in adult male prisoners; however, we confirmed the importance of genetic predisposition for dopaminergic genes (*DBH* and *DRD2*) in aggressive behaviour, memory dysfunction and attention-deficit disorder.

## 1. Introduction

During the last decade, there has been an increased focus on the importance of diet on cognitive ability and behaviour, including aggressive behaviour, impulsivity and attention disorders [1,2,3,4]. It has been suggested that dysfunction in dopaminergic signalling pathways may underlie the development of many of these disorders [5,6,7]. In studies on both adults and children, dopamine (DA) hyperactivity in brain regions linked to reward-related motivation (such as the limbic system and prefrontal cortex) has been associated with increases in impulsive and aggressive behaviours [7,8]. Levels of extracellular DA present in the synaptic cleft can act on dopaminergic receptors, such as dopamine receptors 1 (*DRD1*) and 2 (*DRD2*), which are involved in the development of aggressive behaviour [9,10]. Furthermore, pharmacological blockage of the *DRD2* receptors by antipsychotic treatment revealed an anti-aggression effect in individuals [11]. Animal studies have shown that the selected stimulation of *DRD2* receptors with bromocriptine substantially increased the aggressive behaviour of low-aggression genetically modified mice, whereas the blockade of *DRD2* receptors with sulpiride decreased or prevented the manifestation of aggressivity in high-aggression genetically modified mice, confirming the critical role of *DRD2* receptors in the development of aggressive behaviour [12]. Taking into account the results of a meta-analysis of 24 genetic studies on aggression reporting that heritability accounts for approximately 50% of the variance in aggression, it is not surprising that genes coding for dopaminergic receptors were largely implicated in the genetic vulnerability for aggressive behaviour and impulsivity in several studies [13,14,15,16,17,18]. Both single-nucleotide polymorphisms (SNPs) rs4648317 and rs12364283 in the *DRD2* gene (*DRD2*) were previously associated with impulsive behaviour in a population of healthy volunteers [19]. Functional polymorphism in the *DRD1* gene (encoding for *DRD1*) rs 686 was also reported to be associated with heroin-use and impulsive behaviour [20].

Additional dopaminergic genes encoding for enzymes modulating dopaminergic neurotransmission, such as monoamine oxidase (MAO) and catechol O methyl transferase (COMT) enzymes, play a significant role in the individual response to aggressive behaviour, hyperactivity and attention disorders [13,15,21,22]. Functional SNP in *COMT* (rs 4680), resulting in a valine (Val) to methionine (Met) substitution in the amino-acid protein sequence, leads to differential levels of activity of the COMT enzyme [23,24]. *COMT* rs4680 has been previously associated with aggressive behaviour in both psychiatric and healthy populations [21,25,26]). Similarly, SNP (rs1799836) in the *MAO B* gene was previously reported to be significantly involved in the genetic vulnerability to develop attention-deficit/hyperactivity disorder (ADHD) and negative personality traits in both Han Chinese and Caucasian case-control cohorts [27,28].

Another isoform of the MAO enzyme encoded by *MAO A* has earned the nickname “warrior gene” due to its link to aggression in observational and survey-based studies [29,30]. Metabolic dopamine beta hydroxylase (DBH) is a synaptic enzyme capable of reducing extracellular DA levels. One functional SNP in the promoter (rs1611115) was described as responsible for more than half of DBH enzymatic activity [31]. The minor allele T, leading to a reduction in half of the DBH activity of this genetic variant, was previously found to represent a risk factor for impulsiveness, aggression and adult ADHD [32]. Altogether, dopaminergic gene variants seem to play a critical role in the individual response for impulsive and aggressive behaviours as well as attention and hyperactivity disorders.

Omega-3 long-chain polyunsaturated fatty acids (n-3 LCPUFA) comprise eicosapentaenoic acid (EPA), docosapentaenoic acid (DPA) and docosahexaenoic acid (DHA). The main n-3 LCPUFA involved in neurological development is DHA, particularly at the time of the neural tube closure during very early pregnancy [33] and during the second trimester when the fetus accrues its brain matter [34] but also throughout the lifespan [35]. The Omega-3 Index [36], measured as EPA plus DHA expressed as a percent of total erythrocyte membrane fatty acids, is a good biomarker for the heart tissue levels of n-3 LCPUFA [37]. The Omega-3 Index has been postulated as a new risk factor for death from coronary heart disease [36] and, more recently, for mental health [38]. DHA is critical for all cell membrane structures and functions with particular importance for neuronal cells in the frontal lobes [39]. Evidence suggests that inadequate levels of n-3 LCPUFA can affect the dopaminergic system in the brain and, when combined with genetic and other factors, increase the risk of developing aggression, attention and impulse-control disorders [40].

Taking into account that impulsive aggression is significantly involved in the manifestation of violent and criminal behaviour [5] and the results of our pilot study showing a correlation between high aggressive scoring and a lower Omega-3 Index in adult male prisoners [41], the aim of this study was to investigate the relevant polymorphisms in dopaminergic genes in adult male prisoners in relation to their Omega-3 Index. We hypothesise here that dopaminergic polymorphisms will be associated with aggressive and metacognitive traits and with individual variation with respective Omega-3 Index.

## 2. Materials and Methods

### 2.1. Subjects

One hundred and thirty-six male adult participants were recruited from the South Coast Correctional Centre (SCCC) in Nowra, NSW, Australia, as previously described [41]. Ethics approvals were granted by the University of Wollongong and the Corrective Services (Wollongong, NSW, Australia) ethics committees (11/93185). Of the 136 subjects in this study, 74 (54%) were of Caucasian origin, 14 (10%) of Australian Aboriginal origin, 11 (8%) of Middle Eastern origin, 13 (9.5%) from the Pacific Islands, 12 (8.8%) from the Asian region and 5 (3.6%) of Hispanic origin [41].

### 2.2. Blood Samples and Omega-3 Index

Blood samples were collected in EDTA tubes and subjected to centrifugation at 4 °C for 10 min, and plasma was separated from the packed erythrocytes, which were stored at 80 °C degrees until further use. Levels of EPA and DHA were assessed as previously described [41], and omega-3 levels in erythrocytes were expressed as the Omega-3 Index (calculated as the sum of EPA and DHA expressed as a percent of total erythrocyte fatty acids).

### 2.3. Genotyping

Genomic DNA was isolated from whole blood tubes using a QIAsymphony DSP DNA Midi Kit (96) (Qiagen, Hilden, Germany) following the manufacturer’s recommendations. The tested SNPs (see Table 1) were selected because of the evidence of significant genetic associations with aggressive disorders, memory and/or the relevance of genes in the dopaminergic system. SNP genotyping was performed using the Multiplex MassARRAY^®^ genotyping assay (Sequenom, Inc., San Diego, CA, USA), with the analysis performed by matrix-assisted laser desorption/ionization time-of-flight mass spectrometry (MALDI-TOF MS, Sequenom, Inc., San Diego, CA, USA).

### 2.4. Psychological Assessments

Aggressive behaviours and cognitions were assessed using the Aggression Questionnaire (AQ) [42] and a 7-point observation scale (IBOS: Inmate Behavioural Observation Scale) as rated by prison officers for a duration of four weeks as previously described [41]. Briefly, case notes detailing significant observations and interactions with and between prisoners are routinely digitally recorded by custodial and non-custodial staff. The IBOS is a 7-point scale which classifies these observations of inmate behaviour, creating a numerical rating of frequency and severity of aggression. A weekly total score is derived, enabling longitudinal assessment of behaviour across weeks. A score of −1 is applied to all instances of pro-social behaviour recorded in that week, whereas a score of 0 is given if there were no behaviour of significance recorded. Thereafter, instances of hostile/aggressive behaviour are scored from 1 (non-compliant) to 5 (physically aggressive), with each level of hostility/aggression operationally defined and illustrated by examples [41]. IBOS assessments correlated with the AQ results [41]. The AQ reports five subscales categorised into Physical Aggression, Verbal Aggression, Anger, Hostility and Indirect Aggression.

Since attention-deficit/hyperactivity disorder is one of the most disrupted metacognitive traits in prisoners [43], Brown Attention-Deficit Disorder Scales (BADDS) [44] were administered to this cohort by one of the authors (Byrne, a clinical and forensic psychologist) to assess executive function via measurement of Activation (organising/prioritising), Attention (focussing, sustaining and shifting attention to tasks), Effort (alertness, maintaining effort and speed for processing), Emotion/Affect (managing emotions and frustrations) and Memory (using working memory and recall testing) as previously described [41].

### 2.5. Statistical Analysis

Analyses were performed using SPSS (version 22.0, SPSS Inc., Chicago, IL, USA). The genotypic distribution for the eight tested SNPs was first assessed for deviation from the Hardy–Weinberg equilibrium (HWE) and linkage disequilibrium (LD) using Graphical Overview of Linkage Disequilibrium (GOLD) software [45]. Due to a limited number of participants and, for some SNPs, lower minor allelic frequency, two SNPs in *DRD2* and SNPs in *COMT* and *MAO* A and *B* genes showed genotypic distribution which did deviate from the HWE. Therefore, only rs686 in *DRD1,* rs42744224 in *DRD2* and rs1611115 in *DBH* were further analysed for their association with genotype and Omega-3 Index and AQ and BADDS results using the Kruskal–Wallis test (non-parametric test leading to H value testing analysis of variance between genotypes and assessed behavioural parameters/ Omega-3 Index). When an association was reported as significant between genetic variants and neuropsychological assessments, odd ratios were calculated for the individual genotypes to estimate the risk of returning high scores on the AQ and BADDS for higher/lower total aggression level groups (using a cut-off score of 55 assessed in the AQ questionnaire) and for high/low ADD groups (using a cut off of 40 reported in BADDS). Significance was corrected using the Bonferroni test and set to *p* ≤ 0.025.

## 3. Results

### 3.1. Description and Relation between the Tested Genetic Polymorphisms

Genetic information regarding the SNPs originally considered in this study, as well as the results for the HWE are reported in Table 1. Major allelic frequencies (MAFs) for the tested SNPs were found within the 0–2% range from global MAFs reported in the genetic database, except for the *DRD1* variant being within 13% of the reference allelic frequency [46].

Albeit a strong LD between *DRD2* rs4274224 and rs4581480 (D’ = 0.85) and a moderate LD (ranging between 0.6 and 0.8) between rs4648317 and two other SNPs within the *DRD2* gene, no haplotype block was considered in this study due to the exclusion of genetic markers for violation of the HWE (Figure 1). The demographic features (age and ethnicity) did not show any significant difference when analysed for each of the three SNPs (i.e., rs686, rs4274224 and rs161115) included in this study (*p* > 0.025).

### 3.2. Omega-3 Index, Dopaminergic Variants and Neuropsychological Testing

The levels of Omega-3 Index measured in blood were not associated with any of the *DRD1, DRD2* and *DBH* gene variants (*p* > 0.025; Table 2, Table 3 and Table 4). The *DRD1* variant did not show any genetic association with aggression and metacognitive traits per the AQ (*p* > 0.025; Table 2).

However, *DRD2* rs4274224 was strongly associated with attention-deficit disorder (ADD) (including all subscales of BADDS testing) (H(2)= 11.249, *p* = 0.004) and also with the following individual subscales of BADDS: Activation (H(2) = 9.640, *p* = 0.008), Attention (H (2) = 7.661; 0.022), Effort (H(2) = 12.771, *p* = 0.002) and Memory (H(2) = 9.414, *p* = 0.009; Table 3). The odd ratio (OR) for participants with an AA genotype to return a high score in executive function as assessed by BADDS was 3.12 times more frequent than prisoners with a GG genotype. Only the Affect subscale of BADDS was not found to be significantly associated with the *DRD2* rs4274224 variant, albeit being borderline (H(2) = 7.313, P = 0.026). When considering the genetic association of *DRD2* rs4274224 with the results from the AQ questionnaire, no significant result was reported for both Total Aggression and/or any of the individual subscales on the test (*p* > 0.025, Table 3).

**Table 3 nutrients-14-01379-t003:** *DRD2* genotypes analysis according to Omega-3 Index and neuropsychological testing.

*DRD2* (rs4274224)	AA (*n* = 41)	AG (*n* = 31)	GG (*n* = 30)	*p*
Omega-3 Index	5.2 (4.0, 6.0)	5.1 (4.0, 6.0)	5.6 (4.2, 7.2)	0.591
PhysAgg	22.7 (17.0, 28.0)	20.6 (13.0, 27.5)	22.0 (13.7, 29.2)	0.513
VerbAgg	12.7 (9.0, 15.0)	13.5 (11.0, 17.5)	12.8 (9.5, 16.0)	0.635
Anger	18.0 (14.0, 20.0)	16.7 (11.0, 22.5)	15.2 (10.7, 20.0)	0.256
Hostility	20.1 (14.0; 26.0)	19.8 (13.5, 26.0)	17.0 (12.5, 19.0)	0.220
IndirAgg	14.5 (11.0, 18.0)	14.3 (11.0, 18.5)	12.0 (8.7, 14.0)	0.113
Total ADD	48.9 (32.0, 71.0)	48.0 (23.0, 70.0)	28.9 (13.0, 38.5)	0.004
Activation	11.4 (7.0, 17.0)	10.5 (5.0, 16.5)	7.0 (3.0, 10.0)	0.008
Attention	12.9 (7.0, 18.0)	12.2 (6.0, 19.0)	8.0 (3.0, 12.7)	0.022
Effort	9.0 (4.0, 15.0)	8.8 (4.0, 12.5)	4.5 (1.7, 6.0)	0.002
Affect	7.9 (4.0, 12.0)	8.2 (3.0, 13.0)	5.0 (1.0, 6.0)	0.026
Memory	7.5 (5.0, 11.0)	7.7 (4.0, 12.0)	4.3 (0.7, 6.0)	0.009

In contrast, functional polymorphism *DBH*rs1611115 showed a significant association with the Verbal Aggression subscale from the AQ (H(2) = 7.609, *p* = 0.022; Table 4), with TT genotypes showing an odds ratio (OR) = 4.18 to develop high scores in aggressive behaviour. No association was reported between *DBH*rs1611115 and any other subscale from both the AQ and BADDS tests (*p* > 0.025; Table 4).

**Table 4 nutrients-14-01379-t004:** *DBH* genotypes analysis according to Omega-3 Index and neuropsychological testing.

*DBH* (rs1611115)	CC (*n* = 79)	CT (*n* = 47)	TT (*n* = 6)	*p*
Omega-3 Index	5.1 (3.9, 6.0)	5.6 (4.3, 7.4)	5.0 (4.3, 5.5)	0.275
PhysAgg	20.6 (13.0, 27.0)	23.8 (17.0, 29.0)	17.5 (8.0, 8.0)	0.120
VerbAgg	13.0 (9.0, 16.0)	14.0 (11.0, 17.7)	8.3 (5.0, 7.0)	0.022
Anger	16.5 (11.0, 20.0)	17.5 (13.0, 22.0)	14.6 (7.0, 12.0)	0.521
Hostility	19.5 (14.0, 24.0)	19.4 (13.0, 24.7)	15.5 (8, 18.0)	0.453
IndirAgg	13.9 (10.0, 18.0)	14.0 (12.0, 17.5)	12.0 (10.0, 10.0)	0.589
Total ADD	42.8 (23.0, 65.0)	43.9 (21.2, 62)	65.0 (10.0, 87.0)	0.576
Activation	9.5 (5.0, 14.0)	10.5 (6.0, 15.0)	10.0 (5.0, 22.0)	0.615
Attention	10.9 (6.0, 18.0)	11.9 (5.51, 17.0)	18.3 (3.0, 25.0)	0.427
Effort	7.8 (3.0, 11.0)	7.6 (4.0, 11.0)	12.0 (2.0, 11.0)	0.749
Affect	7.4 (4.0, 12.0)	7.1 (2.2, 11.7)	10.7 (0.0, 15.0)	0.489
Memory	6.9 (3.0, 11.0)	6.7 (2.2, 11.0)	6.7 (0.0, 7.0)	0.983

CC, CT and TT are the genotypes of alleles.

## 4. Discussion

Functional polymorphisms in both *DRD1* and *DBH* genes, along with an intronic variant of *DRD2*, were studied in relation to blood Omega-3 Index and aggressive behavioural and metacognitive assessment in adult male prisoners. None of the tested polymorphisms showed an association with blood Omega-3 Index. However, functional *DBH* polymorphism was significantly associated with Verbal Aggression, whereas *DRD2* polymorphism showed a strong association with total ADD and every subscale of BADDS tests except for Affect (reported borderline to corrected significance set value). In this cohort, no association was observed between functional *DRD1* polymorphism and neuropsychological tested parameters. Interestingly, *DRD1* allelic distribution showed a higher level of A allele in the tested inmate cohort compared with the global minor allelic frequency [46]. Considering that rs686 is a functional SNP located in the promoter region of the *DRD1* gene and its A allele is linked with an increased transcriptional gene activity compared with the G allele [47], a difference in the *DRD1* density number may occur in the tested cohort compared with the general population. Since these receptors mediate aggressive behaviour [9,10], this hypothesis is supported by a previous study reporting higher levels of aggressive behaviour in this cohort of prisoners when compared with university students [43]. The *DRD2* genetic variant did not show any genetic association with a measurement of aggression through the AQ in accordance with a previous study looking at *DRD2* rs1800479’s (located in the downstream of the gene) association with criminal behaviour and self-reported aggression in violent prison inmates [48]. However, significant associations were found in this study between *DRD2* rs4274224 and assessment of executive function through BADDS and total ADD phenotypes. This finding confirms the critical role of intronic *DRD2* rs4274224 on inter-individual variations in personality traits such as negative emotional processing and openness to experience previously reported [49]. The functionality of *DRD2* rs4274224 regulates the neural responses to reward and emotion processing through modulation of dopaminergic neurotransmission in the dorsolateral prefrontal cortex (DLPFC, Brodmann Area BA46/9) and the subgenual cortical area (BA25/32), respectively [49]. Higher-order cognitive and executive functions such as working memory and attention rely on activation of mesocortical dopaminergic pathways, particularly towards the DLPFC [50,51], whereas the dopaminergic neurons in the anterior region of the cingulate cortex (BA25) appear to play a causal role in behavioural changes such as emotion, negative affect and anhedonia [52]. Dysfunction of dopaminergic neurotransmission in these cerebral areas was associated with impaired social and cognitive functions typically seen in neurodevelopmental disorders such as schizophrenia, autism spectrum disorder and ADHD, as well as in depression, impulsivity and substance abuse disorders [53]. The strong association between total ADD, attention and *DRD2* rs4274224 also confirms findings from a previous study investigating the association of this genetic marker with executive functions in an obesity context [54] and from meta-analyses reporting significant associations between some *DRD2* polymorphisms and ADHD [55,56,57].

Another way of modulating dopaminergic neurotransmission without the direct implication of neuronal receptors signalling is to act on the level of extracellular DA in the brain. DBH transforms DA in noradrenaline, allowing modulation of the synaptic levels of DA [31]. Functional polymorphism *DBH* was significantly associated with verbal aggression in the tested cohort of prisoners. Individuals with low enzymatic activity genotypes (CT and TT) are expected to display high levels of DA in the synapse [31]. High levels of DA in the synapse can activate dopaminergic receptors, such as *DRD1* and *DRD2* involved in aggressive behaviour, impulsivity and attention [5,6,7,8]. Functional *DBH* rs1611115 was previously reported to be associated with impulsive personality styles but not with affective disorders [32], suggesting a strong involvement of meso-cortical dopaminergic pathways in this personality trait. This finding is also in line with previous studies reporting that low activity of *DBH* would relate to impulsive behaviours and with higher sensation seeking in adults (consistent with impulsiveness) [58,59]. Verbal aggression involves the activation of higher brain areas (temporal and frontal cortices), whereas physical aggression seems to emerge primarily from the activation of more primitive brain structures such as the limbic system [60]. Consequently, we can hypothesise that the tested DA genetic polymorphisms in this cohort appear to preferentially affect meso-cortical dopaminergic pathways (supporting higher connective function) than mesolimbic dopamine pathways (which involve pleasure and reward) [7]. Further imaging work in the tested cohort will be necessary to validate this hypothesis. Interestingly, differential genetic regulation of DA genes was previously reported through compensatory gene–gene interactions between *COMT* and *DRD2*, with genotypes conferring either elevated prefrontal dopamine or diminished striatal dopamine directly affecting the activation of dopaminergic receptors in the respective brain region [61]. In addition, two mRNA splice variants for *DRD2* were previously reported as the D2 receptor short (DRD2-S) variant mainly found in the pre-synaptic space with relatively greater abundance in the prefrontal cortex compared with the D2 receptor long (DRD2-L) variant located mainly postsynaptically and which is relatively more abundant in the striatum [62,63]. Previous studies reported that genetic differences in the proportion of these two *DRD2* isoforms within the corticostriatal system contribute to symptom variability in schizophrenic patients including working memory disturbances [64,65]. Due to previous associations of *DRD2* splice variants with schizophrenia and its addiction-like phenotypes [62,64,65] and tissue-specific expression [62], it will be interesting to explore this dopaminergic variant in the context of the present study.

Studies have shown a regulation of expression of dopaminergic genes by n-3 LCPUFA [40,66]. DHA, which is an essential structure of the neuronal membrane, can also bind the nuclear receptor retinoid X receptor (RXR). This ligand/receptor complex results in the modulation of gene transcription of dopaminergic genes essential for neuronal development [40]. Knockout RXR mice were found to have abnormalities in synaptic plasticity and learning [67], both neuronal processes underlying memory. A positive association was found between plasma levels of DHA and memory in healthy adults [68], whereas an inverse association between DHA levels and risk of dementia was identified through a meta-analysis [69]. In this study, no significant association was found between Omega-3 Index and dopaminergic polymorphisms (*p* < 0.025). This result may be due to the omega-3 deficiency in this cohort compared with the levels of Omega-3 Index assessed in a university cohort [43] and/or to potential differences in individual metabolic genotypes.

Although there is a growing body of literature showing the beneficial effect of fish-oil consumption (such as a higher level of n-3 LCPUFA in the blood associated with a lower risk of heart disease), high blood levels of n-3 LCPUFA can be detrimental for some individuals depending on their genotype in the context of lipid profiles [70]. A recent study reported that omega-3 supplementation can either aid in reducing blood triglycerides or increase blood triglycerides depending on the genotypes of four variants for the *GJB2*, *SLC12A3*, *ABCA6*, and *MLXIPL* genes in individuals [70]. For instance, individuals with minor allele (G) for *GJB2* rs112803755 polymorphism who received fish oil supplementation showed a decreased level of triglycerides, whereas individuals with the AA genotype who also received the same fish oil supplementation had their triglycerides levels slightly increased [70]. Individual genotypes are critical for the modulation of EPA and DHA in the blood due to their effects on encoding factors and metabolic enzymes (such as fatty acid desaturases and elongases) essential for lipid metabolism and bioactivities [71,72]. Consequently, this genetic variability may contribute to explaining the difference existing between the inmates’ incorporation of EPA and DHA as well as the activation/inhibition of cellular pathways, potentially including dopaminergic pathways [73].

A better understanding of the effects of the Omega-3 Index and genetic variants for dopaminergic genes on aggressive behavioural and executive function phenotypes in adult male prisoners, as well in the general population, will improve prediction of, and response to, aggressive behaviour. Our finding adds to our knowledge regarding the genetic effects on neurobiological mechanisms (such as dopaminergic neurotransmission) and attention and on the aggressive behavioural, attention and memory phenotypes in adult male prisoners. Studies with larger sample sizes should be conducted to confirm these findings and explore the associations from additional genetic polymorphisms in the dopaminergic system and metabolic enzymes with the Omega-3 Index.

## 5. Conclusions

Although n-3 LCPUFA is essential to brain development and maintenance, Omega-3 Index was not significantly associated with genotypes from relevant dopaminergic genes in adult male prisoners. However, as we reported here, the genetic predisposition for dopaminergic genes (*DBH* and *DRD2*) in aggressive behaviour, memory dysfunction and attention deficit disorder has been confirmed.

## Figures and Tables

**Figure 1 nutrients-14-01379-f001:**
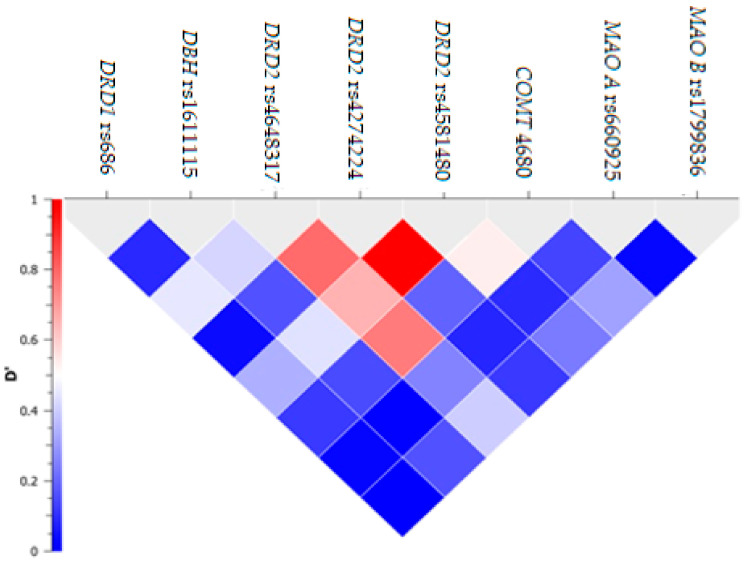
Linkage disequilibrium between selected single-nucleotide polymorphisms (SNPs). On y axis, D’ represents the coefficient of linkage disequilibrium between the tested SNPs (represented on x axis). *DRD1*: dopaminergic receptor D1; *DRD2*: dopaminergic receptor D2; *DBH*: dopaminergic beta hydroxylase; *MAO B*: monoamine oxidase B; *MAO A*: monoamine oxidase A; *COMT*: catechol O methyl transferase.

**Table 1 nutrients-14-01379-t001:** The Hardy–Weinberg equilibrium (HWE) *p*-values and distribution of genotype frequencies of selected single-nucleotide polymorphisms (SNPs).

Genes	Chromosomic Location	SNPs	Alleles	MAF ^1^, (%)	HWE, *p*
*DRD1*	5 (3′UTR Variant)	rs686 *	G > A	29.3	0.66
*DRD2*	11 (Intron Variant)	rs4648317	G > A	89.6	0.02
*DRD2*	11 (Intron Variant)	rs4274224	G > A	54.1	0.42
*DRD2*	11 (Intron Variant)	rs4581480	C > T	87	0.03
*DBH*	9 (3′UTR Variant)	rs1611115 *	C > T	77.7	0.76
*MAO B*	X (Intron Variant)	rs1799836	T > C	53.4	<0.001
*MAO A*	X (Downstream variant 3′)	rs660925	T > C	60.4	<0.001
*COMT*	22 (Missense Variant in exon 4 Val158Met)	rs4680 *	G > A	86.2	<0.001

*DRD1*: dopaminergic receptor D1; *DRD2*: dopaminergic receptor D2; *DBH*: dopaminergic beta hydroxylase; *MAO B*: monoamine oxidase B; *MAO A*: monoamine oxidase A; *COMT*: catechol O methyl transferase; SNPs: single-nucleotide polymorphisms; HWE: Hardy–Weinberg equilibrium, G: Guanine; A: Adenine; T: Thymine; C: Cytosine. ^1^ Major/Reference Allelic Frequency. *** Functional polymorphism.

**Table 2 nutrients-14-01379-t002:** *DRD1* genotypes analysis according to Omega-3 Index and neuropsychological testing.

*DRD1* (rs686)	AA (*n* = 65)	AG (*n* = 51)	GG (*n* = 12)	*p*
Omega-3 Index	5.2 (4.3, 6.0)	5.4 (4.1, 6.7)	5.3 (3.9, 7.0)	0.970
PhysAgg	21.3 (13.0, 26.0)	21.9 (15.0, 29.0)	24.1 (18.2, 27.5)	0.572
VerbAgg	12.9 (9.0, 15.0)	13.3 (11.0, 17.0)	13.6 (11.0, 16.7)	0.875
Anger	16.6 (11.0, 20.0)	17.2 (13.0, 23.0)	17.6 (12.2, 20.5)	0.866
Hostility	19.5 (14.0, 24.0)	18.5 (13.0, 23.0)	21.0 (12.5, 26.5)	0.620
IndirAgg	13.7 (9.0, 17.0)	13.6 (10.0, 17.0)	15.7 (13.2, 18.7)	0.303
Total ADD	47.5 (23.0, 59.0)	42.6 (18.0, 65.0)	52.5 (24.0, 66.2)	0.631
Activation	10.9 (6.0, 16.0)	9.2 (3.0, 14.0)	13.0 (6.2, 16.0)	0.212
Attention	12.2 (4.0, 17.0)	11.6 (5.0, 18.0)	13.1 (7.0, 15.7)	0.873
Effort	8.9 (4.0, 11.0)	7.3 (3.0, 11.0)	10.1 (4.0, 13.0)	0.432
Affect	7.9 (4.0, 11.0)	7.5 (3.0, 12.0)	8.2 (2.0, 12.0)	0.904
Memory	7.3 (3.0, 11.0)	6.6 (2.0, 11.0)	8.1 (4.0, 11.5)	0.731

PhysAgg: Physical Aggression; VerbAgg: Verbal Aggression; IndirAgg: Indirect Aggression; Total ADD: total attention-deficit disorder. AA, AG and GG are the genotypes of alleles.

## Data Availability

Data may be made available upon a reasonable request to the corresponding author.

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
