# Peer review of "Associations between Omega-3 Index, Dopaminergic Genetic Variants and Aggressive and Metacognitive Traits: A Study in Adult Male Prisoners"

_nutrients, 2022, doi:10.3390/nu14071379_

Round 1

Reviewer 1 Report

This is a interesting study regarding the association between EPA and DHA levels and aggressive behavior.

  1. in the methods section please explain who completed the IBOS assessments, when and what the assessment consisted of
  2. please specify who conducted the Brown Attention Deficit Disorder Scales (BADDS) for the inmates
  3. there is a major issue in the statistical test used to test the eight SNPs, given the rarity of the SNPs a different test is needed
  4. study enrolled prisoners in an Australian Corrective Center. I understand that for a study about aggressive behavior that may seem compelling but that may introduce bias. Adding a sample of healthy volunteers should be considered.

Author Response

Reviewer 1

This is a interesting study regarding the association between EPA and DHA levels and aggressive behavior.

  1. in the methods section please explain who completed the IBOS assessments, when and what the assessment consisted of

A: The methods currently states “Aggressive behaviour and cognitions were assessed using the Aggression Questionnaire (AQ) [42] and through a 7 point observation scale (IBOS: Inmate Behavioural Observation Scale) as rated by prison officers for a duration of four weeks as previously described [41]. “ As stated the prison officers completed the IBOS assessments, which has been previously published in reverence 41. We have decided to add a brief description in this manuscript has been done, see lines 140-149.

  1. please specify who conducted the Brown Attention Deficit Disorder Scales (BADDS) for the inmates

A: The prisoners were asked to complete all the BADDS by one of the authors (Byrne, a Clinical and Forensic Psychologist). This has been added to the manuscript (see line 154).

  1. there is a major issue in the statistical test used to test the eight SNPs, given the rarity of the SNPs a different test is needed

A: We can confirm that the tests performed in this study are suitable for the tested genetic markers. All the SNPs had their respective major allelic frequencies (MAFs) within 0-2% range from global MAFs reported in genetic database, except for DRD1 variant being within 13% of the Reference Allelic Frequency, validating the use of these specific SNPs in the selected cohort. In addition, Hardy Weinberg Equilibrium commonly used in genetic study to assess population stratification (Hao & Storey, 2019), each SNP in this cohort leading to the exclusion of four of the tested SNPs. Taking this in account, only the SNPs which did not deviate from HWE were further analysed.

Hao, W., & Storey, J. D. (2019). Extending Tests of Hardy-Weinberg Equilibrium to Structured Populations. Genetics213(3), 759–770. https://doi.org/10.1534/genetics.119.302370

One of our authors Prof Marijka Batterham, is a qualified biostatistician, and she said that there was nothing wrong with the statistical analysis performed in this study.

  1. study enrolled prisoners in an Australian Corrective Center. I understand that for a study about aggressive behavior that may seem compelling but that may introduce bias. Adding a sample of healthy volunteers should be considered.

A: This study was not a comparison of cohorts but rather an investigation within a sample known to exhibit high levels of aggressive behaviour. The presence (or absence) of aggressive behaviour was integral to the nature of relationships between variables. Thus, we do not believe that there was any inherent bias.

Reviewer 2 Report

The study of Fernandez and colleagues correlated specific polymorphisms in dopaminergic genes in adult male prisoners to their Omega-3 Index. The article is well written, and the research appears to be scientifically sound. My only criticism regards the last period of the conclusions (lines 313-16). The authors say that n-3 LCPUFA supplementation could be personalized according to the individual's genetic profile. Still, they did not find any not significant association with genotypes from relevant dopaminergic genes in adult male prisoners. Therefore, the reference to "genetic profile" seems slightly generic, and I suggest eliminating it. 
Moreover, I want to suggest that the author discuss another layer of complexity beyond D2R polymorphism. Indeed, alternative splicing can impact D2R function beyond genetic polymorphism. For example, genetic variability in the proportion of the two alternative D2R mRNA splice variants, D2R-long (D2L) and D2R-short (D2S), influences corticostriatal functioning. In this regard, Colelli et al. (2010) have shown that differences in tissue-specific D2R splicing influence individual variability in addiction-like phenotypes (https://doi.org/10.1111/j.1601-183x.2010.00604.x).
Interestingly, Yamamoto et al. (2021) have found that a schizophrenic patient-specific genetic mutation in heterogeneous nuclear ribonucleoprotein M is involved in the alternative pre-mRNA splicing of DRD2 (https://doi.org/10.1074/jbc.m110.206540).

Author Response

Reviewer 2

The study of Fernandez and colleagues correlated specific polymorphisms in dopaminergic genes in adult male prisoners to their Omega-3 Index. The article is well written, and the research appears to be scientifically sound. My only criticism regards the last period of the conclusions (lines 313-16). The authors say that n-3 LCPUFA supplementation could be personalized according to the individual's genetic profile. Still, they did not find any not significant association with genotypes from relevant dopaminergic genes in adult male prisoners. Therefore, the reference to "genetic profile" seems slightly generic, and I suggest eliminating it. 

A: This part of the conclusion has been deleted.

Moreover, I want to suggest that the author discuss another layer of complexity beyond D2R polymorphism. Indeed, alternative splicing can impact D2R function beyond genetic polymorphism. For example, genetic variability in the proportion of the two alternative D2R mRNA splice variants, D2R-long (D2L) and D2R-short (D2S), influences corticostriatal functioning. In this regard, Colelli et al. (2010) have shown that differences in tissue-specific D2R splicing influence individual variability in addiction-like phenotypes (https://doi.org/10.1111/j.1601-183x.2010.00604.x).
Interestingly, Yamamoto et al. (2021) have found that a schizophrenic patient-specific genetic mutation in heterogeneous nuclear ribonucleoprotein M is involved in the alternative pre-mRNA splicing of DRD2 (https://doi.org/10.1074/jbc.m110.206540).

A: We agree with reviewer 2 that there is an additional genetic layer of variability regarding DRD2 considering DRD2 L and S splice variants. Consequently, we have added the section below in line 295-304:

“In addition, two mRNA spice variants for DRD2 were previously reported as D2 Receptor Short (DRD2-S) variant mainly found in the pre-synaptic space with relatively greater abundance in prefrontal cortex compared to D2 receptor long (DRD2-L) variant being located mainly postsynaptically and relatively more abundant in striatum [62, 63]. Previous studies reported that genetic differences in the proportion of these two DRD2 isoforms within the corticostriatal system contribute to symptoms variability in schizophrenic patients including working memory disturbances [64, 65]. Due to previous associations of DRD2 splice variants with schizophrenia and addiction like phenotypes ([62, 64, 65], and its tissue specific expression [62], it will be interesting to explore this dopaminergic variant in the context of the present study.”

 [62] V. Colelli, M.T. Fiorenza, D. Conversi, C. Orsini, S. Cabib, Strain-specific proportion of the two isoforms of the dopamine D2 receptor in the mouse striatum: associated neural and behavioral phenotypes, Genes Brain Behav 9(7) (2010) 703-11.

[63] B. Giros, P. Sokoloff, M.P. Martres, J.F. Riou, L.J. Emorine, J.C. Schwartz, Alternative splicing directs the expression of two D2 dopamine receptor isoforms, Nature 342(6252) (1989) 923-6.

[64] A. Bertolino, L. Fazio, G. Caforio, G. Blasi, A. Rampino, R. Romano, A. Di Giorgio, P. Taurisano, A. Papp, J. Pinsonneault, D. Wang, M. Nardini, T. Popolizio, W. Sadee, Functional variants of the dopamine receptor D2 gene modulate prefronto-striatal phenotypes in schizophrenia, Brain 132(Pt 2) (2009) 417-25.

[65] K. Yamamoto, T. Kuriu, K. Matsumura, K. Nagayasu, Y. Tsurusaki, N. Miyake, H. Yamamori, Y. Yasuda, M. Fujimoto, M. Fujiwara, M. Baba, K. Kitagawa, T. Takemoto, N. Gotoda-Nishimura, T. Takada, K. Seiriki, A. Hayata-Takano, A. Kasai, Y. Ago, S. Kida, K. Takuma, F. Ono, N. Matsumoto, R. Hashimoto, H. Hashimoto, T. Nakazawa, Multiple alterations in glutamatergic transmission and dopamine D2 receptor splicing in induced pluripotent stem cell-derived neurons from patients with familial schizophrenia, Transl Psychiatry 11(1) (2021) 548.